# Total stochastic gradient algorithms and applications in reinforcement learning

**Paavo Parmas**
Neural Computation Unit
Okinawa Institute of Science and Technology Graduate University
Okinawa, Japan
`paavo.parmas@oist.jp`

## Abstract

Backpropagation and the chain rule of derivatives have been prominent; however, the total derivative rule has not enjoyed the same amount of attention. In this work we show how the total derivative rule leads to an intuitive visual framework for creating gradient estimators on graphical models. In particular, previous "policy gradient theorems" are easily derived. We derive new gradient estimators based on density estimation, as well as a likelihood ratio gradient, which "jumps" to an intermediate node, not directly to the objective function. We evaluate our methods on model-based policy gradient algorithms, achieve good performance, and present evidence towards demystifying the success of the popular PILCO algorithm [5].

## 1 Introduction

A central problem in machine learning is estimating the gradient of the expectation of a random variable with respect to the parameters of the distribution $\frac{\mathrm{d}}{\mathrm{d}\zeta}\mathbb{E}_{x\sim\mathrm{p}(x;\zeta)}\left[\phi(x)\right]$. Some examples include: the gradient of the expected classification error of a model over the data generating distribution, the gradient of the expected evidence lower bound w.r.t. the variational parameters in variational inference [9], or the gradient of the expected reward w.r.t. the policy parameters in reinforcement learning [20]. Usually, such an estimator is needed not just through a single computation, but through a computation graph; a good overview of related problems is given by [18]. Previously, Schulman et al. provided a method to obtain gradient estimators on stochastic computation graphs by differentiating a surrogate loss [18]. While the work provided an elegant method to obtain gradient estimators using automatic differentiation, the resulting *stochastic computation graph* framework has formal rules, which uniquely define one specific type of estimator, and it is not suitable for describing general gradient estimation techniques. For example, determinstic policy gradients [19] or total propagation [14] are not covered by the framework. In contrast, in probabilistic inference, the successful probabilistic graphical model framework [15] only describes the structure of a model, while there are many different choices of algorithms to perform inference. We aim for a similar framework for gradient computation, which we call *probabilistic computation graphs*. Our framework uses the total derivative rule $\frac{\mathrm{d}f}{\mathrm{d}a} = \frac{\partial f}{\partial a} + \frac{\partial f}{\partial b}\frac{\mathrm{d}b}{\mathrm{d}a}$ to decompose the gradient into a sum of partial derivatives along different computational paths, while leaving open the choice of estimator for the partial derivatives. We begin by introducing typical gradient estimators in the literature, then explain our new theorem, novel estimators using a non-standard decomposition of the total derivative, and experimental results.

**Nomenclature** All variables will be considered as column vectors, and gradients are represented as matrices where each row corresponds to one output variable, and each column corresponds to one input variable—this allows applying the chain rule by simple matrix multiplication, i.e. $\frac{\mathrm{d}f(\mathbf{x})}{\mathrm{d}\mathbf{y}} = \frac{\partial f}{\partial \mathbf{x}}\frac{\partial \mathbf{x}}{\partial \mathbf{y}}$. Matrices are vectorised with the vec(*) operator, i.e. $\frac{\mathrm{d}\Sigma}{\mathrm{d}\mathbf{x}}$ means $\frac{\mathrm{dvec}(\Sigma)}{\mathrm{d}\mathbf{x}}$.

## 2 Background: Gradients of expectations

### 2.1 Pathwise derivative estimators

This type of estimator relies on gradients of $\phi$ w.r.t. $\mathbf{x}$, e.g. the Gaussian gradient identities: $\frac{d}{d\mu}\mathbb{E}_{\mathbf{x}\sim\mathcal{N}(\mu,\Sigma)}[\phi(\mathbf{x})] = \mathbb{E}_{\mathbf{x}\sim\mathcal{N}(\mu,\Sigma)}\left[\frac{d\phi(\mathbf{x})}{d\mathbf{x}}\right]$ and $\frac{d}{d\Sigma}\mathbb{E}_{\mathbf{x}\sim\mathcal{N}(\mu,\Sigma)}[\phi(\mathbf{x})] = \frac{1}{2}\mathbb{E}_{\mathbf{x}\sim\mathcal{N}(\mu,\Sigma)}\left[\frac{d^2\phi(\mathbf{x})}{d\mathbf{x}^2}\right]$, cited in [17]. The most prominent type of pathwise derivative estimator are reparameterization (RP) gradients. We focus our discussion on RP gradients, but we mentioned the Gaussian identities to emphasize that RP gradients are not the only possible pathwise estimators, e.g. the derivative w.r.t. $\Sigma$ given above does not correspond to an RP gradient. See [17] for an overview of various options.

**RP gradient for a univariate Gaussian**     To sample from $\mathcal{N}(\mu,\sigma^2)$, sample from a standard normal $\epsilon \sim \mathcal{N}(0,1)$, then transform this: $x = \mu + \sigma\epsilon$. The gradients are $dx/d\mu = 1$ and $dx/d\sigma = \epsilon$. The gradient can then be estimated by sampling: $\frac{d}{d\zeta}\mathbb{E}[\phi(x)] = \mathbb{E}\left[\frac{d\phi(x)}{dx}\frac{dx}{d\zeta}\right]$. For multivariate Gaussians, one can use the Cholesky factor $L$ of $\Sigma = LL^T$ instead of $\sigma$. To differentiate the Cholesky decomposition see [12]. See [17] for other distributions. For a general distribution $p(\mathbf{x};\zeta)$, the RP gradient defines a sampling procedure $\epsilon \sim p(\epsilon)$ and a transformation $\mathbf{x} = f(\zeta,\epsilon)$, which allows moving the derivative inside the expectation $\frac{d}{d\zeta}\mathbb{E}_{\mathbf{x}\sim p(\mathbf{x};\zeta)}[\phi(\mathbf{x})] = \mathbb{E}_{\epsilon\sim p(\epsilon)}\left[\frac{d\phi}{df}\frac{df}{d\zeta}\right]$. The RP gradient allows backpropagating the gradient through sampling operations in a graph. It computes *partial derivatives* through a specific operation.

### 2.2 Jump gradient estimators

We introduce the categorization of *jump gradient estimators*. Unlike pathwise derivatives, which compute local partial derivatives and apply the chain rule through numerous computations, jump gradient estimators can estimate the *total derivative* directly using only local computations—hence the naming: the gradient estimator jumps over multiple nodes in a graph without having to differentiate the nodes inbetween (this will become clearer in later sections in the paper).

**Likelihood ratio estimators (LR)**     Any function $f(\mathbf{x})$ can be stochastically integrated by sampling from an arbitrary distribution $q(\mathbf{x})$: $\int f(\mathbf{x})d\mathbf{x} = \int q(\mathbf{x})\frac{f(\mathbf{x})}{q(\mathbf{x})}d\mathbf{x} = \mathbb{E}_{\mathbf{x}\sim q}[f(\mathbf{x})/q(\mathbf{x})]$. The gradient of an expectation can be written as $\int \phi(\mathbf{x})\frac{dp(\mathbf{x};\zeta)}{d\zeta}d\mathbf{x}$. By picking $q(\mathbf{x}) = p(\mathbf{x})$, and stochastically integrating, one obtains the LR gradient estimator: $\mathbb{E}\left[\frac{dp(\mathbf{x};\zeta)/d\zeta}{p(\mathbf{x};\zeta)}\phi(\mathbf{x})\right]$. One *must* subtract a baseline from the $\phi(\mathbf{x})$ values for this estimator to have acceptable variance: $\mathbb{E}\left[\frac{dp(\mathbf{x};\zeta)/d\zeta}{p(\mathbf{x};\zeta)}(\phi(\mathbf{x}) - b)\right]$. In practice using $b = \mathbb{E}[\phi]$ is a reasonable choice. If $b$ does not depend on the samples, then this leads to an unbiased gradient estimator. Leave-one-out baseline estimates can be performed to achieve an unbiased gradient estimator [11]. Other control variate techniques also exist, and this is an active area of research [7].

In our recent work [14], we introduced the batch importance weighted LR estimator (BIW-LR) and baselines: **BIW-LR:** $\sum_{i=1}^{P}\sum_{j=1}^{P}\left(\frac{dp(\mathbf{x}_j;\zeta_i(\theta))/d\theta}{\sum_{k=1}^{P}p(\mathbf{x}_j;\zeta_k)}(\phi(\mathbf{x}_j) - b_i)\right)/P$, where we use a mixture distribution $q = \sum_i^P p(\mathbf{x};\zeta_i)/P$, and each $\zeta_i$ depends on another set of parameters $\theta$ (in our case the policy parameters), **BIW-Baseline:** $b_i = \left(\sum_{j\neq i}^{P}c_{j,i}\phi(\mathbf{x}_j)\right)/\sum_{j\neq i}^{P}c_{j,i}$, where the importance weights are $c_{j,i} = p(\mathbf{x}_j;\zeta_i)/\sum_{k=1}^{P}p(\mathbf{x}_j;\zeta_k)$.

**Value function based estimators**     Instead of using $\phi(\mathbf{x})$ directly, one can learn an approximator $\hat{\phi}(\mathbf{x})$. The approximator will often require less computational time to evaluate, and could be used for estimating the derivatives. Both LR gradients and pathwise derivatives could be used with evaluations from the approximator. Moreover, it is not necessary to evaluate just one $\mathbf{x}$ point of the estimator, but one could either use a larger number of samples, or try to directly compute the expectation—this leads to a Rao-Blackwellized estimator, which is known to have lower variance. Such estimators have been considered for example in RL in expected sarsa [24, 20] as well as in the stochastic variational inference literature [2, 23], and also in policy gradients [3, 1].

# 3 Total stochastic gradient theorem

Sec. 2 explained how to obtain estimators of the expectation through a single computation, while here we explain how to decompose the gradient of a complicated graph of computations into smaller sections, which can be readily estimated using the methods in Sec. 2. In our framework, we work with the gradient of the marginal distribution. This more general problem directly gives one the gradient of the expectation as well, as the expectation is just a function of the marginal distribution.

## 3.1 Explanation of framework

We define *probabilistic computation graphs* (PCG). The definition is exactly equivalent to the definition of a standard directed graphical model, but it highlights our methods better, and emphasizes our interest in computing gradients, rather than performing inference. The main difference is the explicit inclusion of the *distribution parameters* $\zeta$, e.g. for a Gaussian, the mean $\mu$ and covariance $\Sigma$.

**Definition 1 (Probabilistic computation graph (PCG))** *An acyclic graph with nodes/vertices $V$ and edges $E$, which satisfy the following properties:*

1. *Each node $i \in V$ corresponds to a collection of random variables with marginal joint probability density $\mathrm{p}(\mathbf{x}_i; \zeta_i)$, where $\zeta_i$ are the possibly infinite parameters of the distribution. Note that the parameterization is not unique, and any parameterization is acceptable.*

2. *The probability density at each node is conditionally dependent on the parent nodes: $\mathrm{p}(\mathbf{x}_i|\mathbf{Pa}_i)$ where $\mathbf{Pa}_i$ are the random variables at the direct parents of node $i$.*

3. *The joint probability density satisfies: $\mathrm{p}(\mathbf{x}_1, ..., \mathbf{x}_n) = \prod_{i=1}^n \mathrm{p}(\mathbf{x}_i|\mathbf{Pa}_i)$*

4. *Each $\zeta_i$ is a function of its parents: $\zeta_i = f(\mathbf{Pz}_i)$ where $\mathbf{Pz}_i$ are the distribution parameters at the parents of node i. In particular: $\mathrm{p}(\mathbf{x}_i; \zeta_i) = \int \mathrm{p}(\mathbf{x}_i|\mathbf{Pa}_i)\mathrm{p}(\mathbf{Pa}_i; \mathbf{Pz}_i)\mathrm{d}\mathbf{Pa}_i$*

We emphasize that there is nothing stochastic in our formulation. Each computation is determinstic, although they may be analytically intractable. We also emphasize that this definition does not exclude deterministic nodes, i.e. the distribution at a node may be a Dirac delta distribution (a point mass). Later we will use this formulation to derive stochastic estimates of the gradients.

## 3.2 Derivation of theorem

We are interested in computing the total derivative of the distribution parameters at one node $\zeta_i$ w.r.t. the parameters at another node $\mathrm{d}\zeta_i/\mathrm{d}\zeta_j$, e.g. nodes $i$ and $j$ could correspond to $\phi$ and $\mathbf{x}$ in Sec. 2 respectively. By the total derivative rule: $\frac{\mathrm{d}\zeta_i}{\mathrm{d}\zeta_j} = \sum_{\zeta_m \in \mathbf{Pz}_i} \frac{\partial \zeta_i}{\partial \zeta_m} \frac{\mathrm{d}\zeta_m}{\mathrm{d}\zeta_j}$. Iterating this equation on the $\mathrm{d}\zeta_m/\mathrm{d}\zeta_j$ terms leads to a sum over paths from node $j$ to node $i$:

$$\frac{\mathrm{d}\zeta_i}{\mathrm{d}\zeta_j} = \sum_{Paths(j \to i)} \prod_{Edges(k,l) \in Path} \frac{\partial \zeta_l}{\partial \zeta_k} \tag{1}$$

This equation holds for any deterministic computation graph, and is also well known in e.g. the OJA community [13]. This equation trivially leads to our *total stochastic gradient theorem*, which states that the sum over paths from A to B can be written as a sum over paths from A to intermediate nodes and from the intermediate nodes to B. Fig. 1 provides examples of the paths in Eq. 2 below.

**Theorem 1 (Total stochastic gradient theorem)** *Let $i$ and $j$ be distinct nodes in a probabilistic computation graph, and let $IN$ be any set of intermediate nodes, which block the paths from $j$ to $i$, i.e. $IN$ is such that there does not exist a path from $j$ to $i$, which does not pass through a node in $IN$. We denote $\{a \to b\}$ is the set of paths from $a$ to $b$, and $\{a \to b\}/c$ is the set of paths from $a$ to $b$, where no node along the path except for $b$ is allowed to be in set $c$. Then the total derivative $\mathrm{d}\zeta_i/\mathrm{d}\zeta_j$ can be written with the equation below:*

$$\frac{\mathrm{d}\zeta_i}{\mathrm{d}\zeta_j} = \sum_{m \in IN} \left( \left( \sum_{s \in \{m \to i\}} \prod_{(k,l) \in s} \frac{\partial \zeta_l}{\partial \zeta_k} \right) \left( \sum_{r \in \{j \to m\}/IN} \prod_{(p,t) \in r} \frac{\partial \zeta_t}{\partial \zeta_p} \right) \right) \tag{2}$$

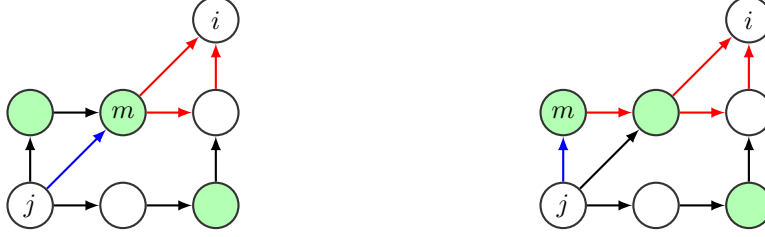

(a) $\{j \to m\}$ paths may not pass through green nodes.    (b) $\{m \to i\}$ paths may pass through green nodes.

Figure 1: Example paths in Equation 2. The green nodes correspond to the intermediate nodes $IN$.

Equations 1 and 2 can be combined to give:

$$\frac{\mathrm{d}\zeta_i}{\mathrm{d}\zeta_j} = \sum_{m \in IN} \left( \left( \frac{\mathrm{d}\zeta_i}{\mathrm{d}\zeta_m} \right) \left( \sum_{r \in \{j \to m\}/IN} \prod_{(p,t) \in r} \frac{\partial \zeta_t}{\partial \zeta_p} \right) \right) \qquad (3)$$

Note that an analogous theorem could be derived by swapping $r \in \{j \to m\}/IN$ and $s \in \{m \to i\}$ with $r \in \{j \to m\}$ and $s \in \{m \to i\}/IN$ respectively. This leads to the equation below:

$$\frac{\mathrm{d}\zeta_i}{\mathrm{d}\zeta_j} = \sum_{m \in IN} \left( \left( \sum_{r \in \{m \to i\}/IN} \prod_{(p,t) \in r} \frac{\partial \zeta_t}{\partial \zeta_p} \right) \left( \frac{\mathrm{d}\zeta_m}{\mathrm{d}\zeta_j} \right) \right) \qquad (4)$$

We will refer to Equations 3 and 4 as the second and first half *total gradient equations* respectively.

### 3.3  Gradient estimation on a graph

Here we clarify one method how the partial derivatives through the nodes $m \in IN$ in the previous section can be estimated. We use the following properties of the estimators in Sec. 2:

- *Pathwise derivative estimators* compute partial derivatives through a single edge, e.g. $\frac{\partial \zeta_m}{\partial \zeta_j}$
- *Jump gradient estimators* sum the gradients across all computational paths between two nodes and directly compute total derivatives, e.g. $\frac{\mathrm{d}\zeta_i}{\mathrm{d}\zeta_m}$

The task is to estimate the derivative of the expectation at a distal node $i$ w.r.t. the parameters at an earlier node $j$: $\frac{\mathrm{d}}{\mathrm{d}\zeta_j} \mathbb{E}_{\mathbf{x}_i \sim \mathrm{p}(\mathbf{x}_i; \zeta_i)} [\mathbf{x}_i]$, through an intermediate node $m$. Note that $\mathbb{E}[\mathbf{x}_i]$ can be picked as one of the distribution parameters in $\zeta_i$. The true $\zeta$ are intractable, so we perform an ancestral sampling based estimate $\hat{\zeta}$, i.e. we sample sequentially from each $\mathrm{p}(\mathbf{x}_* | \mathrm{Pa}_*)$ to get a sample through the whole graph, then $\hat{\zeta}_*$ will simply be the parameters of $\mathrm{p}(\mathbf{x}_* | \mathrm{Pa}_*)$. We refer to one such sample as a *particle*. We use a batch of $P$ such particles $\hat{\zeta}_* = \{\hat{\zeta}_{*,c}\}_c^P$ to obtain a mixture distribution as an approximation to the true distribution. Such a sampling procedure has the properties $\mathrm{p}(\mathbf{x}; \zeta) = \int \mathrm{p}(\mathbf{x}; \hat{\zeta}) \mathrm{p}(\hat{\zeta}) \mathrm{d}\hat{\zeta}$ and $\mathbb{E}_{\mathbf{x}_i \sim \mathrm{p}(\mathbf{x}_i; \zeta_i)} [\mathbf{x}_i] = \mathbb{E}_{\hat{\zeta}_i \sim \mathrm{p}(\hat{\zeta}_i; \zeta_j)} \left[ \mathbb{E}_{\mathbf{x}_i \sim \mathrm{p}(\mathbf{x}_i; \hat{\zeta}_i)} [\mathbf{x}_i] \right]$. For simplicity in the explanation, we further assume that the sampling is reparameterizable, i.e. $\mathrm{p}(\hat{\zeta}_m; \zeta_j) = \int f(\hat{\zeta}_m; \zeta_j, \epsilon_m) \mathrm{p}(\epsilon_m) \mathrm{d}\epsilon_m$. We can write $\frac{\mathrm{d}}{\mathrm{d}\zeta_j} \mathbb{E}_{\hat{\zeta}_i \sim \mathrm{p}(\hat{\zeta}_i; \zeta_j)} \left[ \mathbb{E}_{\mathbf{x}_i \sim \mathrm{p}(\mathbf{x}_i; \hat{\zeta}_i)} [\mathbf{x}_i] \right] = \mathbb{E}_{\epsilon_m \sim \mathrm{p}(\epsilon_m)} \left[ \frac{\partial \hat{\zeta}_m}{\partial \zeta_j} \frac{\mathrm{d}}{\mathrm{d}\hat{\zeta}_m} \mathbb{E}_{\mathbf{x}_i \sim \mathrm{p}(\mathbf{x}_i; \hat{\zeta}_i)} [\mathbf{x}_i] \right]$. The term $\frac{\partial \hat{\zeta}_m}{\partial \zeta_j}$ will be estimated with a pathwise derivative estimator. The remaining term $\frac{\mathrm{d}}{\mathrm{d}\hat{\zeta}_m} \mathbb{E}_{\mathbf{x}_i \sim \mathrm{p}(\mathbf{x}_i; \hat{\zeta}_i)} [\mathbf{x}_i]$ will be estimated with any other estimator, e.g. a jump estimator could be used.

We summarize the procedure for creating gradient estimators from $j$ to $i$ on the whole graph:

1. Choose a set of intermediate nodes $IN$, which block the paths from $j$ to $i$.
2. Construct pathwise derivative estimators from $j$ to the intermediate nodes $IN$.
3. Construct total derivative estimators from $IN$ to $i$, and apply Eq. 3 to combine the gradients.

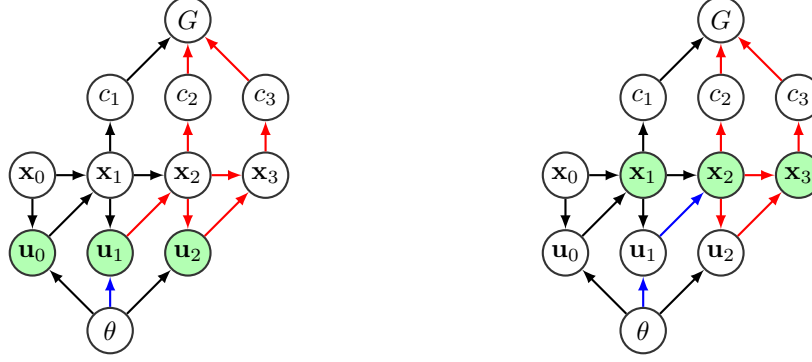

(a) Classical model-free policy gradient      (b) Model-based state-space LR gradient

Figure 2: Probabilistic computation graphs for model-based and model-free LR gradient estimation.

# 4 Relationship to policy gradient theorems

In typical model-free RL problems [20] an agent performs actions $\mathbf{u} \sim \pi(\mathbf{u}_t|\mathbf{x}_t;\theta)$ according to a stochastic policy $\pi$, transitions through states $\mathbf{x}_t$, and obtains costs $c_t$ (or conversely rewards). The agent's goal is to find the policy parameters $\theta$, which optimize the expected return $G = \sum_{t=0}^{H} c_t$ for each episode. The corresponding probabilistic computation graph is provided in Fig. 2a.

In the literature, two "gradient theorems" are widely applied: the policy gradient theorem [21], and the deterministic policy gradient theorem [19]. These two are equivalent in the limit of no noise [19].

**Policy gradient theorem**

$$\frac{\mathrm{d}}{\mathrm{d}\theta}\mathbb{E}\left[G\right] = \mathbb{E}\left[\sum_{t=0}^{H-1} \frac{\mathrm{d}\log\pi(\mathbf{u}_t|\mathbf{x}_t;\theta)}{\mathrm{d}\theta}\hat{Q}_t(\mathbf{u}_t,\mathbf{x}_t)\right] \tag{5}$$

**Deterministic policy gradient theorem**

$$\frac{\mathrm{d}}{\mathrm{d}\theta}\mathbb{E}\left[G\right] = \mathbb{E}\left[\sum_{t=0}^{H-1} \frac{\mathrm{d}\mathbf{u}_t}{\mathrm{d}\theta}\frac{\mathrm{d}\hat{Q}_t(\mathbf{u}_t,\mathbf{x}_t)}{\mathrm{d}\mathbf{u}_t}\right] \tag{6}$$

$\hat{Q}_t$ corresponds to an estimator of the remaining return $\sum_{h=t}^{H-1} c_{h+1}$ from a particular state $\mathbf{x}$ when choosing action $\mathbf{u}$. For Eq. 5 any estimator is acceptable, even a sample based estimate could be used. For Eq. 6, $\hat{Q}$ is usually a differentiable surrogate model. Fig. 2a shows how these two theorems correspond to the same probabilistic computation graph. The intermediate nodes are the actions selected at each time step. The difference lies in the choice of jump estimator to estimate the total derivative following the intermediate nodes—the policy gradient theorem uses an LR gradient, whereas the deterministic policy gradient theorem uses a pathwise derivative to a surrogate model. We believe that the derivation based on a PCG is more intuitive than previous algebraic proofs [21, 19].

# 5 Novel algorithms

In Sec. 3.3 we explained how a particle-based mixture distribution is used for creating gradient estimators. In the following sections, we instead take advantage of these particles to estimate a different parameterization $\Gamma$, directly for the marginal distribution. Although the algorithms have general applicability, to make a concrete example, we explain them in reference to model-based policy gradients using a differentiable model considered in our previous work [14], for which the PCG is given in Fig. 2b. Stochastic value gradients [8], for example, share the same PCG.

## 5.1 Density estimation LR (DEL)

Following the explanation in Sec. 5, one could attempt to estimate the distribution parameters $\Gamma$ from a set of sampled particles, then apply the LR gradient using the estimated distribution $\mathrm{q}(\mathbf{x};\Gamma)$. In

particular, we will approximate the density as a Gaussian by estimating the mean $\hat{\mu} = \sum_i^P \mathbf{x}_i/P$ and variance $\hat{\Sigma} = \sum_i^P (\mathbf{x}_i - \hat{\mu})^2/(P-1)$. Then, using the standard LR trick, one can estimate the gradient $\sum_i^P \frac{\mathrm{d}\log \mathrm{q}(\mathbf{x}_i)}{\mathrm{d}\theta}(G_i - b)$, where $\mathrm{q}(\mathbf{x}) = \mathcal{N}(\hat{\mu}, \hat{\Sigma})$. To use this method, one must compute derivatives of $\hat{\mu}$ and $\hat{\Sigma}$ w.r.t. the particles $\mathbf{x}_i$, then carry the gradient to the policy parameters using the chain rule while differentiating through the model, which is straight-forward. We refer to our new method as the DEL estimator. Importantly, note that while $\mathrm{q}(\mathbf{x})$ is used for estimating the gradient, it is not in any way used for modifying the trajectory sampling.

**Advantages of DEL:** One can use LR gradients even if no noise is injected into the computations.
**Disadvantages of DEL:** The estimator is biased, and density estimation can be difficult.

### 5.2 Gaussian shaping gradient (GS)

Until now, all RL methods have used the second half total gradient equation (Eq. 3). Might one create estimators that use the first half equation (Eq. 4)? Fig.3 gives an example of how this might be done. We propose to estimate the density at $\mathbf{x}_m$ by fitting a Gaussian on the particles. Then $\mathrm{d}\mathbb{E}\left[c_m\right]/\mathrm{d}\Gamma_m$ (the pink edges) will be estimated by sampling from this distribution (or by any other method of integration). This leaves the question of how to estimate $\mathrm{d}\Gamma_m/\mathrm{d}\theta$ (all paths from $\theta$ to $\mathbf{x}_m$). Using the RP method is straight-forward. To use the LR method, we first apply the second half total gradient equation on $\mathrm{d}\Gamma_m/\mathrm{d}\theta$ to obtain terms $\sum_{r\in\{\theta\to x_k\}/IN} \prod_{(p,t)\in r} \frac{\partial\zeta_t}{\partial\zeta_p}$ (blue edges) and $\frac{\mathrm{d}\Gamma_m}{\mathrm{d}\zeta_{x_k}}$ (red edges). In the scenarios we consider, the first of these terms is a single path, and will be estimated using RP. The second term is more interesting, and we will estimate this using an LR method.

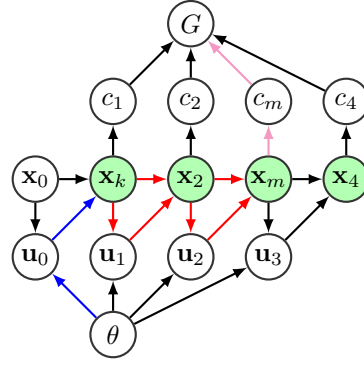

Figure 3: Computational paths in Gaussian shaping gradient

As we are using a Gaussian approximation, the distribution parameters $\Gamma_m$ are the mean and variance of $\mathbf{x}_m$, which can be estimated as $\mu_m = \mathbb{E}\left[\mathbf{x}_m\right]$ and $\Sigma_m = \mathbb{E}\left[\mathbf{x}_m\mathbf{x}_m^T\right] - \mu_m\mu_m^T$. We can obtain LR gradient estimates of these terms $\frac{\mathrm{d}}{\mathrm{d}\zeta_{x_k}}\mathbb{E}\left[\mathbf{x}_m\right] = \mathbb{E}_{\mathbf{x}_k\sim\mathrm{p}(\mathbf{x}_k;\zeta_k)}\left[\frac{\mathrm{d}\log\mathrm{p}(\mathbf{x_k};\zeta_{x_k})}{\mathrm{d}\zeta_{x_k}}(\mathbf{x}_m - \mathbf{b}_\mu)\right]$, $\frac{\mathrm{d}}{\mathrm{d}\zeta_{x_k}}\mathbb{E}\left[\mathbf{x}_m\mathbf{x}_m^T\right] = \mathbb{E}_{\mathbf{x}_k\sim\mathrm{p}(\mathbf{x}_k;\zeta_k)}\left[\frac{\mathrm{d}\log\mathrm{p}(\mathbf{x_k};\zeta_{x_k})}{\mathrm{d}\zeta_{x_k}}(\mathbf{x}_m\mathbf{x}_m^T - \mathbf{b}_\Sigma)\right]$ and $\frac{\mathrm{d}}{\mathrm{d}\zeta_{x_k}}(\mu\mu^T) = 2\mu\frac{\mathrm{d}}{\mathrm{d}\zeta_{x_k}}\mathbb{E}\left[\mathbf{x}_m^T\right]$. In practice, we perform a sampling based estimate $\hat{\zeta}_{x_k}$, and one might be concerned that the estimators are conditional on the sample $\hat{\zeta}_{x_k}$, but we are interested in unconditional estimates. We will explain that the conditional estimate is equivalent. For the variance, note that $\mu_m$ is an estimate of the unconditional mean, so the whole estimate directly corresponds to an estimate of the unconditional variance. For the mean, apply the rule of iterated expectations: $\mathbb{E}_{\mathbf{x}_k\sim\mathrm{p}(\mathbf{x}_k;\zeta_{x_k})}\left[\mathbf{x}_m\right] = \mathbb{E}_{\hat{\zeta}_{x_k}\sim\mathrm{p}(\hat{\zeta}_{x_k})}\left[\mathbb{E}_{\mathbf{x}_k\sim\mathrm{p}(\mathbf{x}_k;\hat{\zeta}_{x_k})}\left[\mathbf{x}_m\right]\right]$ from which it is clear that the conditional gradient estimate is an unbiased estimator for the gradient of the unconditional mean.

**Efficient algorithm for accumulating gradients**  In Fig. 3, for each $\mathbf{x}_k$ node, we want to perform an LR jump to every $\mathbf{x}_m$ node after $k$ and compute a gradient with the Gaussian approximation of the distribution at node $m$. We will accumulate across all nodes during a backwards pass in a backpropagation like manner. Note that for each $k$ and each $m$, we can write the gradient as $\frac{\mathrm{d}\mathbb{E}[c_m]}{\mathrm{d}\Gamma_m}\frac{\mathrm{d}\Gamma_m}{\mathrm{d}\zeta_{x_k}}\left(\frac{\mathrm{d}\zeta_{x_k}}{\mathrm{d}\mathbf{u}_{k-1}}\frac{\mathrm{d}\mathbf{u}_{k-1}}{\mathrm{d}\theta}\right)$. The term $\frac{\mathrm{d}\mathbb{E}[c_m]}{\mathrm{d}\Gamma_m}\frac{\mathrm{d}\Gamma_m}{\mathrm{d}\zeta_{x_k}}$ is estimated as $\frac{\mathrm{d}\mathbb{E}[c_m]}{\mathrm{d}\Gamma_m}\mathbf{z}_m\frac{\mathrm{d}\log\mathrm{p}(\mathbf{x_k};\zeta_{x_k})}{\mathrm{d}\zeta_{x_k}}$, where $\mathbf{z}_m$ corresponds to a vector summarizing the $\mathbf{x}_m - \mathbf{b}_\mu$, etc. terms above. Note that $\frac{\mathrm{d}\mathbb{E}[c_m]}{\mathrm{d}\Gamma_m}\mathbf{z}_m$ is just a scalar quantity $g_m$. We thus use an algorithm which accumulates a sum of all $g$ during a backwards pass, and sums over all $m$ nodes at each $k$ node. See Alg. 1 for a detailed explanation of how it fits together with total propagation [14]. The final algorithm essentially just replaces the usual cost/reward with a modified value, and such an approach would also be applicable in model-free policy gradient algorithms using a stochastic policy and LR gradients.

**Two interpretations of GS**  1. We are making a Gaussian approximation of the marginal distribution at a node. 2. We are performing a type of reward shaping based on the distribution of the

---

**Algorithm 1** Gaussian shaping gradient with total propagation

---

Gaussian shaping gradient for model-based policy search while combining both LR and RP variants using total propagation—an algorithm introduced in our previous work [14].
**Forward pass:** Sample a set of particle trajectories.
**Backward pass:**

**Initialise:** $\frac{\mathrm{d}G_{T+1}}{\mathrm{d}\zeta_{T+1}} = \mathbf{0}$, $\frac{\mathrm{d}J}{\mathrm{d}\theta} = \mathbf{0}$, $G_{T+1} = 0$     ▷ $\zeta$ are the distribution parameters, e.g. all of the $\mu$ and $\sigma$ for each particle

**for** $t = T$ **to** 1 **do**

   $\mu_t = \mathbb{E}\left[\mathbf{x}_t\right]$; $\Sigma_t = \mathbb{E}\left[\mathbf{x}_t\mathbf{x}_t^T\right] - \mu_t\mu_t^T$       ▷ Estimate the marginal distribution as a Gaussian

   **Compute:** $\frac{\mathrm{d}\mathbb{E}[c_t]}{\mathrm{d}\mu_t}$ and $\frac{\mathrm{d}\mathbb{E}[c_t]}{\mathrm{d}\Sigma_t}$, e.g. by sampling from this Gaussian, and using the RP gradient

   **for each** particle $i$ **do**

      $\mathbf{m}_{i,t} = \mathbf{x}_{i,t} - \mu_t$; $\mathbf{v}_{i,t} = \mathrm{vec}\left(\mathbf{x}_{i,t}\mathbf{x}_{i,t}^T - \mathbb{E}\left[\mathbf{x}_t\mathbf{x}_t^T\right]\right)$; $\mathbf{w}_{i,t} = \mathrm{vec}\left(\mathbf{m}_{i,t}\mu_t^T\right)$     ▷ vec($*$) is a vectorization operator which stacks the elements in a matrix/tensor into a column vector

      $g_{i,t} = \frac{\mathrm{d}\mathbb{E}[c_t]}{\mathrm{d}\mu_t}\mathbf{m}_{i,t} + \frac{\mathrm{d}\mathbb{E}[c_t]}{\mathrm{d}\Sigma_t}(\mathbf{v}_{i,t} - 2\mathbf{w}_{i,t})$     ▷ $g$ is a scalar replacing the usual cost/reward

      $G_{i,t} = G_{i,t+1} + g_{i,t}$                ▷ $G$ is the return (the cost of the remaining trajectory)

      $\frac{\mathrm{d}\mathbb{E}[c_t]}{\mathrm{d}\mathbf{x}_{i,t}} = \frac{\mathrm{d}\mathbb{E}[c_t]}{\mathrm{d}\mu_t}\frac{\mathrm{d}\mu_t}{\mathrm{d}\mathbf{x}_{i,t}} + \frac{\mathrm{d}\mathbb{E}[c_t]}{\mathrm{d}\Sigma_t}\frac{\mathrm{d}\Sigma_t}{\mathrm{d}\mathbf{x}_{i,t}}$   ▷ Direct derivative of expected cost for the RP gradient

      $\frac{\mathrm{d}\zeta_{i,t+1}}{\mathrm{d}\mathbf{x}_{i,t}} = \frac{\partial\zeta_{i,t+1}}{\partial\mathbf{x}_{i,t}} + \frac{\mathrm{d}\zeta_{i,t+1}}{\mathrm{d}\mathbf{u}_{i,t}}\frac{\mathrm{d}\mathbf{u}_{i,t}}{\mathrm{d}\mathbf{x}_{i,t}}$

      $\frac{\mathrm{d}G_{i,t}^{RP}}{\mathrm{d}\zeta_{i,t}} = \left(\frac{\mathrm{d}G_{i,t+1}}{\mathrm{d}\zeta_{i,t+1}}\frac{\mathrm{d}\zeta_{i,t+1}}{\mathrm{d}\mathbf{x}_{i,t}} + \frac{\mathrm{d}\mathbb{E}[c_t]}{\mathrm{d}\mathbf{x}_{i,t}}\right)\frac{\mathrm{d}\mathbf{x}_{i,t}}{\mathrm{d}\zeta_{i,t}}$

      $\frac{\mathrm{d}G_{i,t}^{LR}}{\mathrm{d}\zeta_{i,t}} = G_{i,t}\frac{\mathrm{d}\log \mathrm{p}(\mathbf{x}_{i,t})}{\mathrm{d}\zeta_{i,t}}$       ▷ In principle, one could further subtract a baseline from $G$

      $\frac{\mathrm{d}G_{i,t}^{RP}}{\mathrm{d}\theta} = \frac{\mathrm{d}G_{i,t}^{RP}}{\mathrm{d}\zeta_{i,t}}\frac{\mathrm{d}\zeta_{i,t}}{\mathrm{d}\mathbf{u}_{i,t-1}}\frac{\mathrm{d}\mathbf{u}_{i,t-1}}{\mathrm{d}\theta}$

      $\frac{\mathrm{d}G_{i,t}^{LR}}{\mathrm{d}\theta} = \frac{\mathrm{d}G_{i,t}^{LR}}{\mathrm{d}\zeta_{i,t}}\frac{\mathrm{d}\zeta_{i,t}}{\mathrm{d}\mathbf{u}_{i,t-1}}\frac{\mathrm{d}\mathbf{u}_{i,t-1}}{\mathrm{d}\theta}$

   **end for**

   $\sigma_{RP}^2 = \mathrm{trace}(\mathbb{V}\left[\frac{\mathrm{d}G_{i,t}^{RP}}{\mathrm{d}\theta}\right])$; $\sigma_{LR}^2 = \mathrm{trace}(\mathbb{V}\left[\frac{\mathrm{d}G_{i,t}^{LR}}{\mathrm{d}\theta}\right])$     ▷ The sample variance of the particles

   $k_{LR} = 1/\left(1 + \frac{\sigma_{LR}^2}{\sigma_{RP}^2}\right)$                ▷ Weight to combine LR and RP estimators

   $\frac{\mathrm{d}J}{\mathrm{d}\theta} = \frac{\mathrm{d}J}{\mathrm{d}\theta} + k_{LR}\frac{1}{P}\sum_i^P \frac{\mathrm{d}G_{i,t}^{LR}}{\mathrm{d}\theta} + (1 - k_{LR})\frac{1}{P}\sum_i^P \frac{\mathrm{d}G_{i,t}^{RP}}{\mathrm{d}\theta}$     ▷ Combine LR and RP in $\theta$ space

   **for each** particle $i$ **do**

      $\frac{\mathrm{d}G_{i,t}}{\mathrm{d}\zeta_{i,t}} = k_{LR}\frac{\mathrm{d}G_{i,t}^{LR}}{\mathrm{d}\zeta_{i,t}} + (1 - k_{LR})\frac{\mathrm{d}G_{i,t}^{RP}}{\mathrm{d}\zeta_{i,t}}$                ▷ Combine LR and RP in state space

   **end for**

**end for**

---

particles. In particular we are essentially promoting the trajectory distributions to stay unimodal, such that all of the particles concentrate at one "island" of reward rather than splitting the distribution between multiple regions of reward—this may simplify optimization.

## 6   Experiments

We performed model-based RL simulation experiments from the PILCO papers [5, 4]. We tested the cart-pole swing-up and balancing problems to test our GS approach, as well as combinations with total propagation [14]. We also tested the DEL approach on the simpler cart-pole balancing-only-problem to show the feasibility of the idea. We compared particle-based gradients with our new estimators to PILCO. In our previous work [14], we had to change the cost function to obtain reliable results using particles—one of the primary motivations of the current experiments was to match PILCO's results using the same cost as the original PILCO had used (this is explained in greater detail in Section 6.4).

### 6.1   Model-based policy search background

We consider a model-based analogue to the model-free policy search methods introduced in Section 4. The corresponding probabilistic computation graph is given in Fig. 2b. Our notation follows our

previous work [14]. After each episode all of the data is used to learn separate Gaussian process models [16] of each dimension of the dynamics, s.t. $\mathrm{p}(\Delta x_{t+1}^a) = \mathcal{GP}(\tilde{\mathbf{x}}_t)$, where $\tilde{\mathbf{x}} = [\mathbf{x}_t^T, \mathbf{u}_t^T]^T$ and $\mathbf{x} \in \mathbb{R}^D$, $\mathbf{u} \in \mathbb{R}^F$. This model is then used to perform "mental simulations" between the episodes to optimise the policy by gradient descent. We used a squared exponential covariance function $k_a(\tilde{\mathbf{x}}, \tilde{\mathbf{x}}') = s_a^2 \exp(-(\tilde{\mathbf{x}} - \tilde{\mathbf{x}}')^T \Lambda_a^{-1} (\tilde{\mathbf{x}} - \tilde{\mathbf{x}}'))$. We use a Gaussian likelihood function, with noise hyperparameter $\sigma_{n,a}^2$. The hyperparameters, $\{s, \Lambda, \sigma_n\}$ are trained by maximizing the marginal likelihood. The predictions have the form $\mathrm{p}(\mathbf{x}_{t+1}^a) = \mathcal{N}(\mu(\tilde{\mathbf{x}}_t), \sigma_f^2(\tilde{\mathbf{x}}_t) + \sigma_n^2)$, where $\sigma_f^2(\tilde{\mathbf{x}}_t)$ is an uncertainty about the model, and depends on the availability of data in a region of the state-space.

## 6.2 Setup

The cart-pole consists of a cart that can be pushed back and forth, and an attached pole. The state space is $[s, \beta, \dot{s}, \dot{\beta}]$, where $s$ is the cart position and $\beta$ the angle. The control is a force on the cart. The dynamics were the same as in a PILCO paper [4]. The setup follows our prior work [14].

**Common properties in tasks**  The experiments consisted of 1 random episode followed by 15 episodes with a learned policy, where the policy is optimized between episodes. Each episode length was 3s, with a 10Hz control frequency. Each task was evaluated separately 100 times with different random number seeds to test repeatability. The random number seeds were shared across different algorithms. Each episode was evaluated 30 times, and the cost was averaged, but note that this was done only for evaluation purposes—the algorithms only had access to 1 episode. The policy was optimized using an RMSprop-like learning rule [22] from our previous work [14], which normalizes the gradients using the sample variance of the gradients from different particles. In the model-based policy optimization, we performed 600 gradient steps using 300 particles for each policy gradient evaluation. The learning rate and momentum parameters were $\alpha = 5 \times 10^{-4}$, $\gamma = 0.9$ respectively—the same as in our previous work. The output from the policy was saturated by $\mathrm{sat}(u) = 9\sin(u)/8 + \sin(3u)/8$, where $u = \tilde{\pi}(\mathbf{x})$. The policy $\tilde{\pi}$ was a radial basis function network (a sum of Gaussians) with 50 basis functions and a total of 254 parameters. The cost functions were of the type $1 - \exp(-(\mathbf{x} - \mathbf{t})^T Q(\mathbf{x} - \mathbf{t}))$, where $\mathbf{t}$ is the target. We considered two types of cost functions: 1) *Angle Cost*, a cost where $Q = \mathrm{diag}([1, 1, 0, 0])$ is a diagonal matrix, 2) *Tip Cost*, a cost from the original PILCO papers, which depends on the distance of the tip of the pendulum to the position of the tip when it is balanced. These cost functions are conceptually different—with the *Tip Cost* the pendulum could be swung up from either direction, with the *Angle Cost* there is only one correct direction. The base observation noise levels were $\sigma_s = 0.01$ m, $\sigma_\beta = 1$ deg, $\sigma_{\dot{s}} = 0.1$ m/s, $\sigma_{\dot{\beta}} = 10$ deg/s, and these were modified with a multiplier $k \in \{10^{-2}, 1\}$, such that $\sigma^2 = k\sigma_{base}^2$.

**Cart-pole swing-up and balancing**  In this task the pendulum starts hanging downwards, and must be swung up and balanced. We took some results from our previous work [14]: PILCO; reparameterization gradients (RP); Gaussian resampling (GR); batch importance weighted LR, with a batch importance weighted baseline (LR); total propagation combining BIW-LR and RP (TP). We compared to the new methods: Gaussian shaping gradients using the BIW-LR component (GLR), Gaussian shaping gradients combining BIW-LR and RP variants using total propagation (GTP). Moreover, we tested GTP when the model noise variance was multiplied by 25 (GTP+$\sigma_n$).

**Cart-pole balancing with DEL estimator**  This task is much simpler—the pole starts upright and must be balanced. The experiment was devised to show that DEL is feasible and may be useful if further developed. The *Angle Cost* and the base noise level were used.

## 6.3 Results

The results are presented in Table 1 and in Fig. 4. Similarly to our previous work [14], with low noise, methods which include LR components do not work well. However, the GTP+$\sigma_n$ experiments show that injecting more noise into the model predictions can solve the problem. The main important result is that GTP matches PILCO in the *Tip Cost* scenarios. In our previous work [14], one of the concerns was that TP had not matched PILCO in this scenario. Looking only at the costs in Fig. 4b and 4c does not adequately display the difference. In contrast, the success rates show that TP did not perform as well. The success rates were measured both by a threshold which was calibrated in previous work (final loss below 15) as well as by visually classifying all experimental runs. Both methods agreed.

Table 1: Success rate of learning cart-pole swing-up

| Cost func. | $\sigma_o^2$ multiplier | PILCO | RP | GR | LR | TP | GTP | GLR | GTP+$\sigma_n$ |
|---|---|---|---|---|---|---|---|---|---|
| Angle Cost | $k = 10^{-2}$ | **0.88** | 0.69 | 0.63 | 0.57 | **0.82** | 0.65 | 0.42 | **0.88** |
| Angle Cost | $k = 1$ | 0.79 | 0.74 | 0.89 | **0.96** | **0.99** | **0.9** | **0.93** | |
| Tip Cost | $k = 10^{-2}$ | **0.92** | 0.44 | 0.47 | 0.36 | 0.54 | 0.6 | 0.45 | 0.8 |
| Tip Cost | $k = 1$ | **0.73** | 0.15 | **0.68** | 0.28 | 0.48 | **0.69** | 0.35 | |

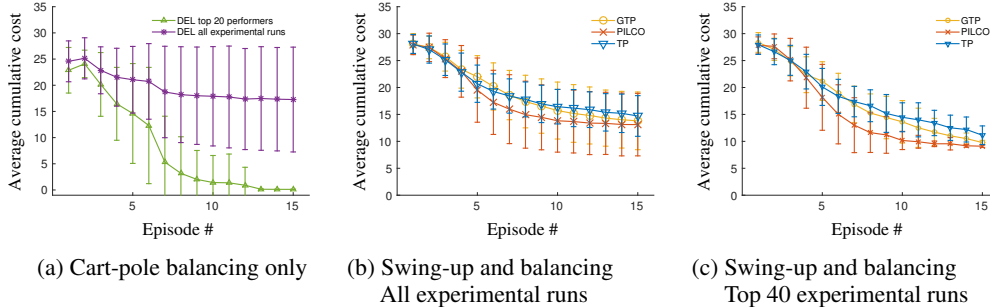

(a) Cart-pole balancing only    (b) Swing-up and balancing All experimental runs    (c) Swing-up and balancing Top 40 experimental runs

Figure 4: Data-efficiency and performance of learning algorithms on cart-pole tasks. Figures 4b and 4c correspond to the $k = 1$, *Tip Cost* case.

The losses of the peak performers at the final episode were    TP: $11.14 \pm 1.73$,    GTP: $9.78 \pm 0.40$, PILCO: $9.10 \pm 0.22$, which also show that TP was significantly worse. While the peak performers were still improving, the remaining experiments had converged. PILCO still appears slightly more data-efficient; however, the difference has little practical significance as the required amount of data is low. Also note that in Fig. 4b TP has smaller variance. The larger variance of GTP and PILCO is caused by outliers with a large loss. These outliers converged to a local minimum, which takes advantage of the tail of the Gaussian approximation of the state distribution—this contrasts with prior suggestions that PILCO performs exploration using the tail of the Gaussian [5].

## 6.4 Discussion

Our work demystifies the factors which contributed to the success of PILCO. It was previously suggested that the Gaussian approximations in PILCO smooth the reward, and cause unimodal trajectory distributions, simplifying the optimization problem [10, 6]. In our previous work [14], we showed that the main advantage was actually that it prevents the curse of chaos/exploding gradients. In the current work we decoupled the gradient and reward effects, and provided evidence that both factors contributed to the success of Gaussian distributions. While GR often has similar performance to GTP, there is an important conceptual difference: GR performs resampling, hence the trajectory distribution is not an estimate of the true trajectory distribution. Moreover, unlike resampling, GTP does not remove the temporal dependence in particles, which may be important in some applications.

## 7 Conclusions & future work

We have created an intuitive graphical framework for visualizing and deriving gradient estimators in a graph of probabilistic computations. Our method provides new insights towards previous policy gradient theorems in the literature. We derived new gradient estimators based on density estimation (DEL), as well as based on the idea to perform a *jump estimation* to an intermediate node, not directly to the expected cost (GS). The DEL estimator needs to be further developed, but it has good conceptual properties as it should not suffer from the curse of chaos nor does it require injecting noise into computations. The GS estimator allows differentiating through discrete computations in a manner that will still allow backpropagating pathwise derivatives. Finally, we provided additional evidence towards demystifying the success of the popular PILCO algorithm. We hope that our work could lead towards new automatic gradient estimation software frameworks which are not only concerned with computational speed, but also the accuracy of the estimated gradients.

**Acknowledgments**

We thank the anonymous reviewers for useful comments. This work was supported by OIST Graduate School funding and by JSPS KAKENHI Grant Number JP16H06563 and JP16K21738.

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
