[Reviews · NeurIPS 2018]

Reviewer 1



This paper provides another formalism for gradient estimation in probabilistic computation graphs. Using pathwise derivative and likelihood ratio estimators, existing and well-known policy gradient theorems are cast into the proposed formalism. This intuition is then used to propose two new methods for gradient estimation that can be used in a model-based RL framework. Some results are shown that demonstrate comparable results to PILCO on the cart-pole task. Quality: the idea in this work is interesting, and the proposed framework and methods may prove useful in RL settings. However, as the authors themselves state in line 251, "DEL is feasible and may be useful if further developed". It seems that this further development is necessary to truly evaluate the potential benefits of the method. In particular, how would the proposed gradient estimation methods scale to more complex, high-dimensional tasks, such as real robots? If the methods are difficult to scale, is there some other insight that can be derived from the formalism that can help? Clarity: the paper is generally well-written with minor grammatical mistakes. The material is technically dense and will likely require careful reading for most readers, but at a glance the math seems to check out. The relationships to other methods is useful for situating the work within the larger body of prior work. Originality: the paper proposes a framework analogous to, but more general than, stochastic computation graphs, and uses this framework to derive novel gradient estimation methods. In this sense, the work is fairly novel. Significance: as stated earlier, further development of the proposed methods is necessary to evaluate the significance of this work, and without this development, it is unclear whether the proposed methods and their benefits warrant acceptance. ------ Update after reading the other reviews and author response and discussing with the other reviewers: I have bumped up my original score from a 3 to a 5. The main reason for this is that, as the authors point out, the DEL estimator is only a small part of what they wish to present, and I unnecessarily focused too much attention on this part. But I do not necessarily agree with the rest of the rebuttal. The authors mention stochastic computation graphs from NIPS '15 and how they derived known gradient estimators as opposed to new estimators. However, I see this as a strength of the paper, not a weakness, because the known estimators were already shown to be useful in practice and unifying them under a single framework was useful for tying theory together with practice simply and elegantly. Are there any gradient estimators used in practice that can be explained with the proposed PCG framework but not with previous frameworks such as stochastic computation graphs? This would certainly be interesting, and I did not see this in the paper or the rebuttal, though I may have missed it. The authors also mention that GS allows for better understanding the success of PILCO. But after reading the PIPPS paper, I do not think that this paper does much to further explain PILCO's strong empirical performance. Section 6.4 could perhaps be rewritten to make this more clear, but parts of this still seem speculative. Is the claim that both PILCO and GS-based methods are successful due to the Gaussian approximations? I'm not entirely sure what strong evidence backs this claim while eliminating other possibilities, e.g., is there evidence that the trajectory distributions indeed stay more unimodal for these approaches compared to the other approaches? In summary, I agree with the points that the contributions could be valuable, and for that reason in particular I think that the paper should be thoroughly revised, the experiments fleshed out more concretely, and the speculations addressed empirically. I think that by doing so, the authors could potentially arrive at a truly impactful contribution.

Reviewer 2



This article deals with the problem of estimating the gradient of the expectation of a function of a random variable whose distribution is parameterized, an important problem in machine learning. It briefly reviews existing approaches (mainly reparametrization and likelihood ratio) and introduces the notion of probabilistic computation graph (PCG), as well as a way to compute partial derivatives in it based on the total derivative rule. Based on this, the authors discuss how to use this to estimate the gradient on such a graph. A relationship to policy gradient in reinforcement learning (RL) is then discussed. Novel algorithms for this specific case are presented, and experimented on variations of the cart-pole problem, with comparison to PILCO (the likely motivation for this paper). I think this paper makes valuable contributions, but it is pretty hard to follow. Besides heavily relying on [1] (which is also quite hard to follow), the link between the proposed PCG and the original problem (estimating the gradient of the expectation of a function of a random variable whose distribution is parameterized) is unclear. It becomes clearer with the relationship to RL, but I wonder what more generality it could have. The proposed algorithms are also unclear, and it is hard to draw conclusions from the experiments. More detailed comments, in the reading order: * bold p and p are not the same, but look very similar (especially as indices), would be better to choose another notation * l.61-65: how is BIW-LR linked to the original problem? We have one xi, then many xis, this is not motivated nor explained * l.70 (and other places): what means that a computation can be performed locally? * PCG: could explain briefly how it differs from a stochastic computation graph (even if clearly not the same thing). More importantly, how is this linked to the original problem? It really looks like the problem changes between 2.1 and 3, without being explained or motivated. For example, how computing the quantity in Eq.1-4 helps solving the original problem? * Eqs 1-4: would be good to give a hint of how these different equations are useful (somehow clearer after, but much later) * Sec. 3.3 is not clear. The link to the original problem is still unclear. There is no longer the phi function in the expectations, it disappeared. Why the true zera are intractable, and how to choose the hat zeta is also unclear. Therefore, the following is also unclear. How to choose the nodes IN could be explained. * Sec. 5 is not clear. In 5.1, how are sampled the particles, in what space, and why? What density is approximated as a Gaussian and why? Same problem in sec. 5.2, what are the xs, according to what distribution is mu the expectation, and so on. * Sec. 6 is hard to follow, even knowing PILCO and having been through [1]. It is also hard to draw conclusions about the proposed approach, what does it brings? * l.192, about the accuracy of the estimated gradients, are there any clues that the proposed approach improves accuracy? ======= Thanks for the rebuttal. I think that this paper has valuable contributions, but also that the required revision regarding clarity is too significant. So, I'm still on the weak reject side. Whatever the outcome, I think that the paper could have broader impact and audience with a strong revision.

Reviewer 3



In this paper, the authors investigate the application of a total gradient theorem to the evaluation of gradient of expectations in probabilistic computation graphs. By combining the total gradient rule with other known gradient estimators (reparametrization and score function estimator), they derive new estimators, in particular in the context of policy gradient for reinforcement learning. They use these new estimators to explain the empirical success of PILCO. Though I found the exposition a bit lacking, I found the paper overall very interesting (the Gaussian shaping gradient in particular); both because the problem of estimating gradients of expectations is fundamental (and therefore new estimators are highly valuable) and for the careful analysis in the experimental section. - I felt that what the authors really wanted to present was derivative of marginal distribution with respect to other marginal distributions, but because this would have been mathematically cumbersome, introduced 'virtual' parameters \xi, even though the distribution for which \xi are parameters are never actually specified, and it in the vast majority of PCGs, there would be no known distribution with parameters \xi corresponding to the true marginal distribution. This leads to a total stochastic gradient theorem that takes a strange form, since it involves variables which are both underspecified and never actually computed. I understand why this was done, but this makes this entire section read a bit strangely. - Theorem 1 - even if it follows from the total derivative rule, deserves a short proof (and the total derivative rule, a reference). - The second and third section don't tie very well together, since one section (section 3) relates gradient of distribution parameters to others, and the other (section 2), is instead about computing gradients of expectations. Of course, the parameters of a distribution can often be written as expectation of sufficient statistics (a fact used by the Gaussian Shaping gradient), but this is never made clear and the sections are not connected enough. For that matter, it's not clear how does the policy gradient (eq 5) actually relates to a total gradient theorem (what is the critic a gradient of?). The connection between (5) and (6) being different estimators of the same computation graph was already known (see for instance [2]). The connection between path-relative partial gradient and reparametrization is also only implicit in the text. - Section 5.2 was confusingly written. As I understand, the first half total gradient theorem is applied with variables x_m, and the first part of the gradient is estimated by making a gaussian assumption and noting that the parameters of the marginal gaussian can be written as expectation of sufficient statistics; at which point the score function estimator is applied to compute the expectation gradient. - For the PCG framework, aren't properties 1 and 2 implying properties 3 and 4? Minor: - I didn't feel the introduction of a new framework was necessarily warranted- as stochastic computation graphs were also introduced as factor graphs and the problem of interest (gradient of expectation) was the same. The SCG framework did not explicitly call for sampling the graph forward, though it is true that the methods introduced call for sampling - but practically, so do the methods in this paper. Both frameworks define distributions over variables, identify the relevant problem as an intractable deterministic distribution, and neither paper actually offers fully deterministic computation on those distributions as practical algorithms. The main difference is the explicit focus on computing marginal distributions in the PCG framework. - Given you give variables meaningful names in the PCG for model-free RL, it would be simpler overall not to index the parameters variables \xi, but perhaps to use a notation like \xi(x), \xi(u), etc. \xi(x_m) would be much easier to parse than \xi_m, since that notation could refer to the sufficient statistics of any m-indexed variable. - Hunch: the gaussian shaping gradient could be related to policy gradients for average-cost MDPs [1], since in both cases , the gradient of the expected value is written as gradient of marginal distributions times rewards (instead of gradient of conditional distribution times returns in the classical PG theorem). The settings are different and no Gaussian approximation is made, but this felt related. [1] Simulation-Based Optimization of Markov Reward Processes, Marbach and Tsitsiklis. [2] Stochastic Value Gradients, Heess et. Al